# Exploring the Influence of Gut–Brain Axis Modulation on Cognitive Health: A Comprehensive Review of Prebiotics, Probiotics, and Symbiotics

**DOI:** 10.3390/nu16060789

**Published:** 2024-03-10

**Authors:** Mónika Fekete, Andrea Lehoczki, Dávid Major, Vince Fazekas-Pongor, Tamás Csípő, Stefano Tarantini, Zoltán Csizmadia, János Tamás Varga

**Affiliations:** 1Department of Public Health, Faculty of Medicine, Semmelweis University, 1089 Budapest, Hungary; fekete.monika@med.semmelweis-univ.hu (M.F.); ceglediandi@freemail.hu (A.L.); major.david@semmelweis.hu (D.M.); pongor.vince@semmelweis.hu (V.F.-P.); csipo.tamas@med.semmelweis-univ.hu (T.C.); stefano-tarantini@ouhsc.edu (S.T.); 2National Institute for Haematology and Infectious Diseases, Department of Haematology and Stem Cell Transplantation, South Pest Central Hospital, 1097 Budapest, Hungary; 3Department of Neurosurgery, The University of Oklahoma Health Sciences Center, Oklahoma City, OK 73104, USA; 4Department of Health Promotion Sciences, College of Public Health, The University of Oklahoma Health Sciences Center, Oklahoma City, OK 73104, USA; 5Peggy and Charles Stephenson Oklahoma Cancer Center, Oklahoma City, OK 73104, USA; 6Faculty of Health Sciences, University of Pécs, 7621 Pécs, Hungary; penituki@gmail.com; 7Department of Pulmonology, Semmelweis University, 1083 Budapest, Hungary

**Keywords:** cognitive function, prebiotic, probiotic, symbiotic, dementia, randomized controlled trial

## Abstract

Recent research exploring the relationship between the gut and the brain suggests that the condition of the gut microbiota can influence cognitive health. A well-balanced gut microbiota may help reduce inflammation, which is linked to neurodegenerative conditions. Prebiotics, probiotics, and symbiotics are nutritional supplements and functional food components associated with gastrointestinal well-being. The bidirectional communication of the gut–brain axis is essential for maintaining homeostasis, with pre-, pro-, and symbiotics potentially affecting various cognitive functions such as attention, perception, and memory. Numerous studies have consistently shown that incorporating pre-, pro-, and symbiotics into a healthy diet can lead to improvements in cognitive functions and mood. Maintaining a healthy gut microbiota can support optimal cognitive function, which is crucial for disease prevention in our fast-paced, Westernized society. Our results indicate cognitive benefits in healthy older individuals with probiotic supplementation but not in healthy older individuals who have good and adequate levels of physical activity. Additionally, it appears that there are cognitive benefits in patients with mild cognitive impairment and Alzheimer’s disease, while mixed results seem to arise in younger and healthier individuals. However, it is important to acknowledge that individual responses may vary, and the use of these dietary supplements should be tailored to each individual’s unique health circumstances and needs.

## 1. Introduction

The dietary habits and lifestyle of the Western civilized world have a significant impact on the gut microbiota, influencing cognitive functions [1,2,3]. Diets rich in added sugars and refined carbohydrates, with low fiber content and limited probiotic intake, can adversely affect the gut microbiota, potentially correlating with cognitive issues [1,4]. The prevalence of artificial additives, the consumption of pre-packaged, ultra-processed foods, and high-saturated fat diets can negatively influence both the gut microbiota and cognitive health and may be linked to cardiovascular problems as well [2]. The excessive use of antibiotics and other medications can harm beneficial gut bacteria, leading to long-term negative effects on cognitive functions [3]. Sedentary lifestyles, low levels of physical activity, and chronic stress can also detrimentally impact cognitive performance [5]. Therefore, maintaining and supporting a healthy gut microbiota can contribute to optimizing cognitive functions and play a crucial role in maintaining the balance between the brain and the gut.

As the aging of society poses significant challenges to healthcare systems and societies, it becomes increasingly important to recognize the interplay between gut health and cognitive well-being, especially in older populations. Estimates suggest that by 2030, one in six members of the global population will be aged 60 or older [6]. Furthermore, the number of individuals aged 60 and above is expected to double by 2050, potentially reaching 2.1 billion. Notably, the age group over 80 years old may triple between 2020 and 2050, reaching up to 426 million [6]. The World Health Organization (WHO) emphasizes several factors to preserve cognitive functions and support health in the elderly, including maintaining a balanced diet rich in vitamins, antioxidants, and nutrients; sustaining regular physical activity; avoiding smoking; limiting alcohol consumption; preventing social isolation; and maintaining social connections. All these factors are crucial for emotional and cognitive well-being [7].

The nourishment of the gut microbiota has long involved the use of probiotic products containing various beneficial strains, as well as fermented foods and strains that specifically stimulate proper brain function through gut–brain axis stimulation [8]. According to the WHO’s definition, probiotics are “live microorganisms which, when administered in adequate amounts, confer a health benefit on the host” [9]. Probiotic dairy products contain beneficial bacteria (mainly lactobacilli and bifidobacteria) in sufficient quantities, which can potentially create a less hospitable environment for pathogens in the colon lumen, thereby hypothetically supporting the integrity of the gut flora and the immune system [10]. Another nutritional factor is dietary fibers acting as prebiotics (food for probiotic bacterial strains), as the diverse consumption of dietary fibers also promotes microbiome diversity [11]. They stimulate the growth and activity of beneficial bacteria, promote a healthy balance of the gut flora, and support digestion. Concurrently intaking probiotics and prebiotics (symbiotics) can promote a healthy gut flora balance and have strengthening effects on the immune and digestive systems [12]. Increasing research results confirm that supporting the gut flora with probiotics and strengthening the gut–brain axis can offer a new treatment alternative for those with mental health issues such as major depression, anxiety disorders, chronic fatigue syndrome, attention deficit hyperactivity disorder (ADHD), depression caused by irritable bowel syndrome (IBS), mood disorders, and stress-induced harm [13,14]. Moreover, even in healthy individuals exposed to increased stress, their positive effects on the brain and psychological well-being are evident, suggesting that their use is safe and necessary for everyone, especially in the context of Western nutrition and modern urban lifestyles [15,16]. Therefore, the aim of our comprehensive review is to examine the publications of the last 5 years, with a particular focus on the administration of probiotics, prebiotics, and symbiotics and the cognitive outcomes derived from their use.

## 2. Methods

We conducted a comprehensive literature search using PubMed, ClinicalTrials.gov, and Cochrane Central Register of Controlled Trials (CENTRAL) databases from 31 January 2019 to 31 January 2024. Our focus was on randomized controlled trials (RCTs) and human clinical trials exploring dietary supplement interventions and their correlation with cognitive function. Specific and MESH keywords such as “probiotic”, “prebiotic”, “symbiotic”, “cognition”, “memory”, “executive function”, “dementia”, “mild cognitive impairment”, “Alzheimer’s disease”, “cognition disorder”, “randomized controlled trial”, and “controlled clinical trial” were used, without language restrictions. The search utilized conjunctions like “AND” or “OR” between keywords. After removing indexed duplicate articles, we screened titles and abstracts, excluding those that did not meet our inclusion criteria. The selected articles underwent careful evaluation based on their full texts. The goal of this review was to provide an up-to-date overview of the relationship between probiotics, prebiotics, symbiotics, and cognitive function, following the PICO (Population, Intervention, Comparison, and Outcomes) criteria. Table 1 outlines the inclusion and exclusion criteria, while Figure 1 illustrates a flowchart of the article selection process. In total, this review includes 23 articles, comprising 837 healthy individuals, 539 patients with mild cognitive impairment (MCI), and 299 patients with Alzheimer’s disease (AD). The current review article is a continuation of our previous study [17], which focused on specialized dietary supplements in the context of aging. Since probiotics/prebiotics are food components that can also be found in fermented foods such as live yogurt cultures, kefir, fermented soybean and probiotic fermented milk, our current summary study includes not only dietary supplement capsules.

## 3. Results

### 3.1. The Impact of Probiotic/Prebiotic/Symbiotic Supplementation on Cognitive Function in Individuals without Cognitive Impairment

Pro- and prebiotics, as well as symbiotics, influence the gut–brain axis, enhance the functioning of the central nervous system, and play a crucial role in the prevention and treatment of various conditions such as dementia, depression, attention deficit hyperactivity disorder, autism, chronic fatigue syndrome, etc. However, their impact is not limited to diseases; it is also noteworthy in healthy individuals. The supplementation of these products improves sleep quality through the modulation of the gut flora, plays a significant role in maintaining hormonal balance and the balanced functioning of the nervous system, and even contributes to shaping overall well-being and mood. In our review of the literature, we found that probiotic/prebiotic/symbiotic supplementation, administered for a minimum of 4 weeks (up to a maximum of 6 months), resulted in a significant improvement (*p* < 0.05) in cognitive performance among healthy aging individuals (over 60 years old) (see Table 2). This improvement was observed in various domains, including memory, attention, and visuospatial/constructional abilities [18,19,20,21,22,23,24,25,26]. However, one study [27] did not find a significant improvement in cognitive performance due to probiotic supplementation, specifically in seniors (average age 64.3 years) who were physically active and met recommended exercise guidelines. In healthy young individuals (under 30 years), cognitive performance did not change significantly after 8 weeks of pre–probiotic supplementation [28,29]. Although, a study with a low sample size (*n* = 26) involving healthy young adults (25–45 years), conducted by Cannavale CN et al. [30], reported a significant improvement in relational memory after the participants consumed a fermented probiotic beverage for 4 weeks (*p* < 0.05).

### 3.2. The Influence of Probiotic/Prebiotic/Symbiotic Supplementation on Cognitive Function in Individuals Diagnosed with Mild Cognitive Impairment

In our analysis, six RCTs [31,32,33,34,35,36] were carefully chosen, revealing a significant improvement in cognitive functions, including improvements in MMSE and Japanese version of Alzheimer’s Disease Assessment Scale (ADAS-Jcog) scores, in both male and female subjects (see Table 3). For mild cognitive impairment (MCI) patients aged over 60, a regimen of probiotics like Lactobacillus, Bifidobacterium, and a prebiotic (inulin) was administered for 2–6 months, once or twice a day, with a dosage of ×10^10^ CFU/day. Asaoka et al.’s study [32] highlighted that Bifidobacterium breve MCC1274 administration enhanced cognitive function in individuals with MCI, showing an improvement in scores on specific neuropsychological test subscales, especially ADAS-Jcog and MMSE. Additionally, the 24-week probiotic supplementation effectively slowed the progression of brain atrophy, evaluated through the voxel-based specific regional analysis system for Alzheimer’s disease (VSRAD) based on brain magnetic resonance imaging (MRI). Xiao J et al.’s [33] groundbreaking double-blind, placebo-controlled human trial demonstrated the cognitive enhancement benefits of Bifidobacterium breve A1 in suspected MCI individuals. Both primary (Repeatable Battery for the Assessment of Neuropsychological Status; RBANS) and secondary endpoints (Japanese version of the MCI Screen; JMCIS) were successfully achieved after 16 weeks of consistent consumption. The treatment demonstrated excellent tolerability, with no side effects being reported. The findings unveiled a significant enhancement in cognitive functions among participants receiving B. breve A1 compared to the placebo group. Notably, the RBANS score exhibited a remarkable 11.3-point improvement with B. breve A1 compared to the placebo (*p* < 0.0001). Improvements were also noted in RBANS domain scores, including immediate memory (*p* < 0.0001), visuospatial/constructional (*p* < 0.0001), and delayed memory (*p* < 0.0001). Fei Y et al.’s study [34] demonstrated the effectiveness of probiotic interventions in restoring various symptoms in MCI, indicating improvements in multiple neural behaviors, enhanced sleep quality, and the alleviation of recorded gastrointestinal symptoms through probiotic supplementation. Kobayashi et al. [35] observed positive outcomes after administering probiotics to elderly individuals with MCI for 24 weeks, showing that the intervention favorable effects on MMSE scores, suggesting an amelioration of cognitive impairment and a potential reduction in the risk of dementia. Hence, dietary supplementation with probiotics and prebiotics emerges as a novel, accessible therapeutic option for treating or preventing MCI without side effects.

### 3.3. The Effect of Probiotic Supplementation on Cognitive Function in Individuals Afflicted by Alzheimer’s Disease

In our comprehensive research, we delved into four studies, including three RCTs and one clinical study, investigating the impact of probiotic supplementation on cognitive functions in elderly patients with Alzheimer’s disease. The average follow-up period across these studies was 12 weeks. One study involved patients receiving a probiotic fermented dairy beverage at a dosage of 2 mL/kg/day [37], while three other studies administered the probiotic in capsule form, with a dosage of twice 10^15^ CFU/day [37,38,39,40]. Additionally, one study included selenium supplementation at a dose of 200 μg/day [40]. The outcomes revealed a significant improvement in the measured cognitive functions, particularly in memory and attention, as evident in the MMSE test (see Table 4). Ton AMM et al.’s study [37] provided an initial assessment of the positive impacts of kefir supplementation over 90 days on cognitive function, along with the biomarkers associated with systemic oxidative stress, inflammation, and cell damage in elderly individuals with Alzheimer’s disease. Akhgarjand et al.’s trial [38], focusing on probiotic supplementation’s influence on cognitive status in patients with mild and moderate Alzheimer’s disease, demonstrated a noteworthy enhancement in MMSE total score. Moreover, improvements were observed in the categorical verbal fluency test (CFT), Activities of Daily Living (AD), and Generalized Anxiety Disorder (GAD-7) in response to probiotic supplementation. Kobayashi Y et al. [39] explored the effects of a 12-week supplementation with Bifidobacterium breve in elderly Japanese patients experiencing difficulties with their memory, reporting a significant increase in both Repeatable Battery for the Assessment of Neuropsychological Status (RBANS) scores and MMSE total scores. Tamtaji et al.’s study [40] showcased that the co-supplementation of probiotics and selenium over a 12-week period in Alzheimer’s disease patients had positive effects on MMSE score; hs-CRP; serum total antioxidant capacity; total glutathione; markers of insulin metabolism; triglycerides; and VLDL-, LDL-, and total/HDL cholesterol ratios. However, it did not impact other biomarkers associated with inflammation and oxidative stress.

## 4. Discussion

Probiotics, prebiotics, and symbiotics support the health of the digestive system and have gained increasing attention recently for their positive effects on cognitive functions and dementia prevention. Various studies [41,42,43] and our current review suggest that probiotics may help and slow down the decline in cognitive abilities in older age. Prebiotics can facilitate the growth and activity of probiotics, contributing to the health of the gut flora, which may play a role in the functioning of the immune and nervous systems [44]. Symbiotics may also offer additional benefits for both gut flora and cognitive health [45]. In conclusion, our review indicates that there is a close connection between the gut microbiome and cognitive aging. Through pursuing further investigations, we could realize the practical applications of theoretical findings. The more evidence we accumulate, the more likely targeted interventions will slow the decline in cognitive abilities in older age.

The microbiome–gut–brain axis is a complex system that describes interactions between the brain, the gut, and the microbiome [4,46,47,48]. There are numerous ways in which the microbiome and the brain interact, partially realized through direct neuronal, hormonal, and immune pathways [49,50]. During aging, the composition of the microbiome may change, influencing cognitive health [51,52]. As time progresses, microbiome diversity may decrease, potentially due to aging processes and cognitive decline [4,53,54]. Additionally, inflammatory processes may become more prevalent during aging [55], potentially harming brain structures and functions, indicating a bidirectional relationship. The question arises: how do probiotics exert their effects? Among several possible mechanisms, the production of neurotransmitters such as γ-aminobutyric acid (GABA), serotonin, catecholamines, and acetylcholine is a key consideration [56]. These compounds can directly impact neural activity, cognitive functions, mood, and emotional well-being. Another effect on neurological performance occurs through the stress response system, specifically via the inhibition of the hypothalamic–pituitary–adrenal (HPA) axis and cortisol synthesis. Researchers believe that irregular functioning of the HPA axis may underlie mood disorders and cognitive problems [57]. The third essential point is that gut microorganisms participate in regulating the immune system’s function [58]. Therefore, probiotics have anti-inflammatory effects, and inflammatory responses can influence the functioning of neurons in the brain and cognitive functions [59]. It is worth noting that gut microorganisms also contribute to the production of certain vitamins and minerals, which are crucial for the brain and cognitive functions [17,60].

According to numerous researchers, a close association can be observed between chronic inflammation, elevated levels of inflammatory mediators, factors supporting inflammation in the body, and the development of cognitive and/or mood disorders [61,62,63,64]. Susceptibility to inflammation in the body can be heightened by abdominal obesity (metabolically active adipose tissue) [65] and the phenomenon of Leaky Gut Syndrome [66], attributed to the impaired functioning of the intestinal mucosal barrier and its underlying microbiome. Disruption of the cohesive microbial layer ensuring the integrity and healthy functioning of the intestinal mucosa increases the vulnerability of the mucosa and enhances the risk of developing Leaky Gut Syndrome. The essence of this negative process involves the opening of junctional structures between intestinal mucosal cells, allowing protein fragments, bacteria, or bacterial-derived substances to enter the circulation. These circulating substances trigger an immune response, leading to sterile inflammation in tissues, enhancing the production of inflammatory factors (resulting in subclinical inflammation at the systemic level), and potentially serving as the basis for autoimmune processes [67]. It is likely that chronic low-grade sterile inflammation also plays a role in age-related alterations in the cerebral microcirculation, contributing to the pathogenesis of vascular cognitive impairment [68,69,70,71,72,73,74,75,76,77,78,79]. Probiotics play a crucial role in directly inhibiting inflammation threatening the nervous system and may prevent the circulation of substances that activate the immune system [42]. In summary, when consumed in the form of beneficial strains as probiotics, these microorganisms positively influence brain function, enhance cognitive functions, improve overall well-being, and contribute to a healthier life in terms of emotion and mood regulation for both healthy individuals and those suffering from cognitive disorders. Additionally, they enhance resilience against stress [14,18]. Studies indicate their positive effects on the brain and cognitive functions, even in fundamentally healthy individuals exposed to increased stress [80,81]. When applying probiotics, attention should be given to dosage and duration; ideally, they should be taken for a minimum of 4 weeks at a dosage of at least 10 billion CFUs per day [82].

In the case of MCI, cognitive, attentional, and memory functions are no longer as proficient as in healthy aging, but the decline has not yet reached the level of dementia. MCI can be considered a precursor to dementia, where solving complex tasks and understanding written information become challenging [83]. The prevalence of MCI is 6.7% at the age of 60–64 years, 8.4% at 65–69 years, 10.1% at 70–74 years, 14.8% at 75–79 years, and 25.2% at 80–84 years. The significance of this condition lies in the fact that within 2 years, dementia develops in 14.9% of individuals affected by MCI [84]. Currently, there is no known medication capable of preventing the progression of MCI into dementia. According to the guidelines of the American Academy of Neurology (AAN), physical activity and exercise are primarily encouraged in MCI cases, in contrast to pharmacological therapy [84]. Cognitive training, establishing proper sleep hygiene, managing potential depression, and improving quality of life are measures that can also alleviate MCI symptoms [84,85]. An important task is to identify other modifiable risk factors. Existing results suggest that probiotic treatment has a positive impact on cognitive abilities in individuals with MCI [86,87]. In our current study, the most commonly used bacterial strains were various types of Lactobacillus and Bifidobacterium breve. Patients received these probiotics for varying durations, ranging from 8 to 24 weeks. All six RCT studies showed significant results, indicating that individuals consuming probiotics achieved significantly better outcomes on tests such as the Mini-Mental State Examination. Overall, the consumption of probiotics had a positive effect on the cognitive abilities of individuals with MCI.

The development of Alzheimer’s disease [69,88,89,90,91,92,93,94,95,96,97,98,99,100,101,102,103,104,105,106,107] and the mechanisms through which the microbiome can contribute to the progression of Alzheimer’s disease have been investigated [108,109,110,111,112,113,114,115,116,117,118,119,120,121,122,123,124,125,126,127,128,129,130,131,132,133,134,135,136,137,138]. Numerous studies are also available regarding treatments with probiotics [139,140,141,142]. In one study [143], it was observed that the consumption of a combined probiotic preparation (*Lactobacillus acidophilus*, *Lactobacillus casei*, *Bifidobacterium bifidum*, and *Lactobacillus fermentum*) for 12 weeks had a significantly positive impact on the cognitive abilities of the study participants in the Mini-Mental State Examination compared to the control group. Our current research reinforces these findings [38,39,40]. Results from a meta-analysis analyzing 12 published studies [144] suggest that adhering more closely to the Mediterranean diet is associated with a lower likelihood of dementia, Alzheimer’s disease, and overall cognitive decline. A more recent meta-analysis examining 34,168 participants [145] reached a similar conclusion: individuals following a plant-based Mediterranean diet have a 21% lower likelihood of developing cognitive impairments and a 40% lower likelihood of developing Alzheimer’s disease. This study also highlights positive changes in the gut microbiota among individuals following a plant-based diet, which can be attributed to a diet rich in probiotics and fiber. According to this study, at least one-third of individuals with Alzheimer’s disease may have developed the condition due to lifestyle-related factors [145]. Therefore, it will be crucial in the future to explore other lifestyle factors that may influence the state of the gut microbiota, for which we currently lack information, as well as to understand the specific mechanisms through which the gut microbiome affects cognitive functions.

While research studies on probiotics and neurodegenerative diseases are still ongoing and interest in them is increasing, another research topic has emerged in the literature, namely the concept of “para-probiotics” and “postbiotics”, which play an important role in understanding the benefits of fermented foods even after cooking [146]. This research topic focuses on the concept of “functional food”, which refers to foods that can provide additional health benefits beyond basic nutrition. Para-probiotics are microorganisms that mimic probiotic effects but do not survive in the gastrointestinal tract. These substances can function similarly to probiotics, as they can improve gut flora composition or support the immune system, but they do not remain present in the gut for as long as traditional probiotics. On the other hand, postbiotics are substances produced during digestion, such as short-chain fatty acids, which can also contribute to maintaining a healthy gut flora.

Tempeh is a fermented soy product traditionally of Indonesian origin that is becoming increasingly popular worldwide among those following vegetarian and vegan diets. Tempeh can be considered a functional food because it is rich in protein, fiber, and other nutrients, as well as probiotics and bioactive compounds such as isoflavonoids. Research indicates that consuming tempeh may offer several health benefits, including reducing the risk of cardiovascular diseases, normalizing blood lipid levels, reducing the risk of diabetes, supporting bone health, and even reducing the risk of cancer. Furthermore, due to its high protein content, tempeh may be beneficial for individuals following vegetarian and vegan diets who need to ensure adequate protein intake. In terms of future prospects, further research is needed to better understand the relationship between tempeh and human health, as well as its effects on various health conditions and populations. Additionally, further developing tempeh and increasing its consumption could contribute to diversifying our diets and promoting healthier lifestyles [146].

In the context of preventing healthy aging and dementia, as well as preserving cognitive functions, several factors play a crucial role. Regular physical activity and maintaining a healthy diet and optimal weight are key elements in preserving healthy aging and cognitive functions [7]. In addition, maintaining social relationships and social interactions can contribute to brain health and the preservation of cognitive functions [147]. Engaging in mental challenges such as learning, solving puzzles, reading, creative activities, and intellectual stimulation can help to maintain and enhance cognitive functions [148]. Adequate sleep plays a significant role in brain regeneration and the maintenance of cognitive functions. Chronic sleep deprivation may increase the risk of dementia [149]. Optimal stress management and maintaining emotional well-being, along with maintaining a healthy cardiovascular system, are also important, as good blood circulation contributes to the brain receiving an adequate supply of oxygen [150]. Diabetes and high blood sugar levels are associated with the development of dementia [151], emphasizing the importance of monitoring blood sugar levels and adopting a balanced diet [152]. These factors collectively contribute to preserving health and cognitive functions in older age. However, it is important to note that each individual is unique, and genetic factors may also play a role in cognitive health and the development of dementia. Nevertheless, adopting a healthy lifestyle and preventive measures can offer a wide range of benefits.

Promoting healthy aging is a prioritized concern, given the rapidly aging population in Europe and the rest of the world [153]. Research areas focusing on the potential positive effects of this process on cognitive functions and the favorable regulatory impact on the gut microbiota, including the investigation of supplementation with vitamins, antioxidants, and omega-3 fatty acids, are of paramount importance [17,154]. This summary study provides substantial evidence supporting the use of probiotics as an alternative strategy for promoting cognitive health in aging. Daily supplementation with probiotics may have beneficial effects on cognitive functions such as memory and attention in both young and elderly individuals, whether healthy or ill [29,155]. This summary underscores the critical importance and urgency of using probiotics and advancing research related to cognitive aging. Ongoing studies aim to deepen our understanding of the interactions within the microbiome–gut–brain axis, particularly exploring mechanisms between the gastrointestinal tract and the nervous system [122,156,157,158,159,160,161,162,163,164,165,166,167,168,169,170,171]. Future research should focus on more precise analyses and explorations of the composition of gut microbiota, identifying which strains are dominantly associated with conditions such as anxiety, depression, Parkinson’s disease, and other psychiatric and cognitive disorders, including Alzheimer’s disease [172,173,174,175,176,177,178,179,180,181,182,183,184]. Areas that remain less explored, such as the various negative associations of different bacterial strains, including inflammatory diseases, neuroinflammation, and metabolic disorders or postpartum depression, require further investigation for a deeper understanding. It is crucial to emphasize that probiotics are not miracle cures, and supporting healthy aging requires other fundamental factors, including a healthy diet, regular exercise, and an adequate quantity and quality of sleep, all of which contribute to overall health and well-being.

## 5. Limitations

Among the limitations, it is crucial to mention the small sample sizes in the studies (some less than 50 participants), which could have been a significant confounding factor in our research. Additionally, some studies only administered probiotics for a short duration (4 weeks). In the studies, the exclusion of the intake of other dietary supplements (e.g., vitamins, antioxidants, omega-3 supplements) was not addressed, nor was there any inquiry about antibiotic use. The studies did not exclude the application of various diets such as vegan, Mediterranean, Dietary Approaches to Stopping Hypertension (DASH Diet), Mediterranean–DASH Intervention for Neurodegenerative Delay (MIND diet), etc., which directly influence gut flora composition and may affect cognitive functions. Another essential confounding factor in cognitive performance is the level of physical activity, which was also not considered in the included studies, with one exception. Furthermore, the examined studies utilized different cognitive measurement scales, making a comparison between them impossible. They also did not exclude other mental activities or the use of herbs, coenzymes, micronutrients, etc., in their research. Although numerous research plans exist, few studies have been completed, and no precise dosage regimen has been defined for various types and stages of dementias with the identification of disease-specific probiotic strains; this needs to be determined in future research. In addition, future research should incorporate prospective studies to investigate whether the long-term use of probiotic supplements can contribute to preventing cognitive decline.

## 6. Conclusions

Based on our findings, it can be stated that the supplementation of probiotics, prebiotics, and symbiotics improves cognitive performance in both healthy individuals and those with cognitive disorders (e.g., MCI/AD), even after regular intake for just 1–2 months. However, for the confirmation of these results, further long-term clinical studies are necessary to gain a more precise understanding of the neuroprotective effects of these dietary supplements. It will also be crucial to determine how patients in different stages of various degenerative conditions respond to probiotic supplementation at different dosages during the progression of the disease. Therefore, additional long-term randomized controlled trials are required.

## Figures and Tables

**Figure 1 nutrients-16-00789-f001:**
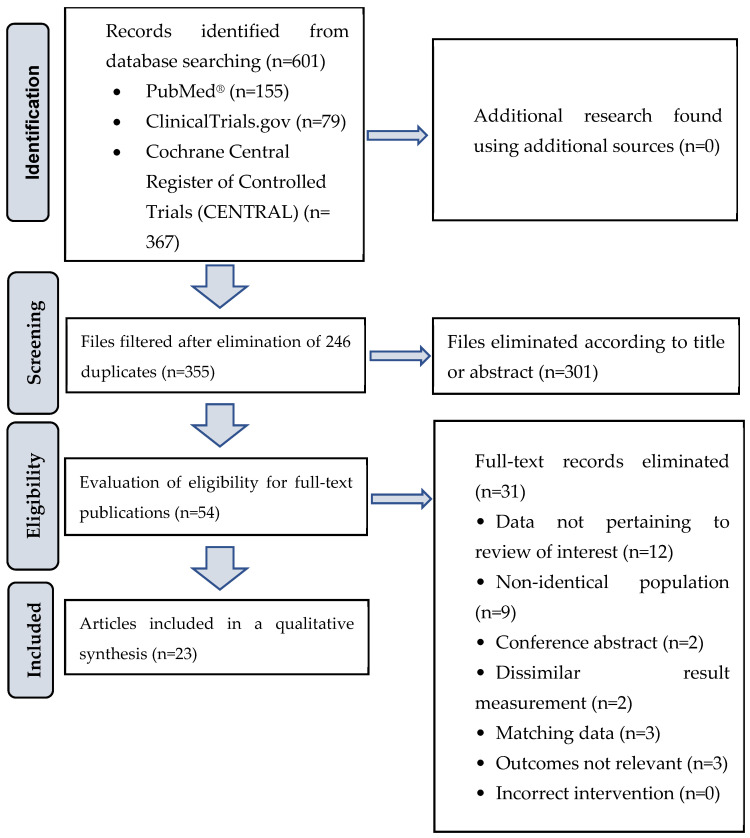
Flow diagram of article selection process.

**Table 1 nutrients-16-00789-t001:** Inclusion and exclusion criteria for studies in the review.

Inclusion Criteria	Description
Study design	Randomized controlled trial or human clinical trial
Study population	Individuals in good health or patients admitted with a diagnosis of mild cognitive impairment or Alzheimer’s disease
Intervention	Prebiotic, probiotic, and symbiotic interventions
Language of publication	No limitations on language
Published articles	In the PubMed, ClinicalTrials.gov, and Cochrane Central Register of Controlled Trials (CENTRAL) databases
Output concepts	Score representing cognitive performance and various assessments of cognitive functions: attention, calculation, memory, verbal fluency, psychomotor speed, visual-constructional ability, neuropsychological function, reaction time, and psychocognitive tests. Various cognitive functions and their assessment tools include validated questionnaires like the Mini-Mental State Examination, Verbal Fluency Test, Repeatable Battery for the Assessment of Neuropsychological Status, Rapid Visual Information Processing, Wisconsin Card Sorting Test, Japanese version of Alzheimer’s Disease Assessment Scale, etc.
**Exclusion Criteria**	
In vitro studies	
Animal experiments	
Interventions targeting a range of health conditions, including but not limited to malignancies, post-traumatic stress disorder, depression, anxiety, stroke, multiple sclerosis, chronic cerebral ischemia, polycystic kidney disease, schizophrenia, bipolar disorder, autism spectrum disorder, attention deficit hyperactivity disorder, diabetes mellitus, fibromyalgia, hepatic encephalopathy, perioperative and postoperative conditions, Huntington’s disease, cirrhosis hepatis, allergic rhinitis, frailty syndrome, psychosis, mood disorders, bipolar disorder, epilepsy, portal hypertension, and a human immunodeficiency virus (HIV)-positive status	
Interventions tailored for various life stages and situations, including but not limited to premature, infancy, adolescence, pregnancy, and interventions designed for athletes	
Nutritional guidance, dietary recommendations, and food interventions	
Brief interventions lasting less than 4 weeks	
Weight loss support for overweight patients	

**Table 2 nutrients-16-00789-t002:** The effects of probiotic/prebiotic/symbiotic supplementation on cognitive functions in healthy subjects.

Study	Design	Mean Follow-Up	Country	Sample Size	Average Age (Year)	Sex Male/Female (%)	Intervention	Main Results
Kim CS et al. [18]	RCT	12 weeks	Republic of Korea	63	71.5 ± 4.3	49.06/50.94	The research participants were administered a probiotic supplement consisting of a total of 1 × 10^9^ CFU of *Bifidobacterium bifidum* BGN4 and *Bifidobacterium longum* BORI, suspended in soybean oil.	At week 12, individuals in the probiotics group demonstrated a notable enhancement in mental flexibility compared to those in the placebo group (*p* < 0.05). Enhanced cognitive and mental capabilities following the administration of probiotic supplements.
Shi S et al. [19]	RCT	8 weeks	China	60	64.1 ± 3.4	57/43	Participants were required to consume one sachet of probiotics (BB68S, 5 × 10^10^ CFU per sachet) or a placebo daily.	BB68S demonstrated a significant enhancement in participants’ cognitive functions, as evidenced by a notable 18.89-point increase in the total RBANS score post-intervention (*p* < 0.0001). This improvement was particularly prominent in the domains of immediate memory, visuospatial/constructional abilities, attention, and delayed memory.
Sakurai K et al. [20]	RCT	12 weeks	Japan	78	76.8 ± 4.6	46/54	The participants received a 1 g packet containing *Lactiplantibacillus plantarum* OLL2712 cells in a quantity exceeding 5 × 10^9^ daily.	The analysis results indicated that the intake of OLL2712 exerted a protective effect on memory function in the elderly (*p* < 0.05).
Czajeczny D et al. [21]	RCT	6 weeks	Poland	38	19–31	100% female	The individuals received a probiotic supplement containing *Bifidobacterium lactis* BS01 and *Lactobacillus acidophilus* LA02.	In the group supplemented with probiotics, there was a significant improvement in cognitive performance compared to the placebo group, as assessed by the Wisconsin Card Sorting Test (WCST) (*p* < 0.05).
M Ni [22]	RCT	12 weeks	United Kingdom	72	>60	37/63	The elderly subjects received a daily intake of one sachet of prebiotic dietary supplement.	In comparison to the placebo group, the prebiotic intervention arm exhibited an enhanced cognition factor score (0.482; 95% CI 0.823–0.141; *p* = 0.014).
Azuma N et al. [23]	RCT	12 weeks	Japan	80	64.6 ± 7.1	50/50	During the 12-week study period, participants ingested test drinks containing 1 × 10^10^ CFU of GCL2505 per 100 g along with 2.0 g of inulin per 100 g.	Substantial enhancements were observed in the scores within the neurocognitive index domain (*p* = 0.027), evaluating overall cognitive function, as well as across the attention, cognitive flexibility, and executive function domains (*p* = 0.044).
Berding K et al. [24]	RCT	4 weeks	Ireland	18	26 ± 1.3	100% female	The female participants received 12.5 g of Litesse^®^ Ultra (>90% polydextrose (PDX) polymer).	PDX demonstrated enhanced cognitive flexibility, indicated by a reduction in errors during the Intra–Extra Dimensional Set Shift (IED) task. Improved sustained attention was evident through a higher number of correct responses and rejections in the Rapid Visual Information Processing (RVP) task.
Sanborn V et al. [25]	RCT	3 months	USA	145	64.3 ± 5.5	40.7/59.3	The intervention involved Culturelle Vegetarian Capsules which contained a blend of 10 billion CFUs of *Lactobacillus rhamnosus* GG for the experimental group.	Probiotic supplementation with Lactobacillus rhamnosus GG was linked to enhanced cognitive performance in middle-aged and older adults (*p* < 0.05).
Louzada ER et al. [26]	RCT	6 months	Brazil	49	77.2 ± 1.3	80/20	The synbiotic group was administered two daily doses (6 g + 6 g) of a compound containing fructooligosaccharide (6 g), *L. paracasei* (10^9^ CFU), *L. rhamnosus* (10^9^ CFU), *L. acidophilus* (10^9^ CFU), and *B. lactis* (10^9^ CFU).	According to their results, the supplement exhibits modest effects on reducing depressive symptoms and more favorable effects on cognitive functions in elderly individuals (MMSE; *p* < 0.05).
Sanborn V et al. [27]	RCT	8 weeks	USA	127	64.3 ± 3.6	42/58	The probiotic supplement for the subjects was *Lactobacillus rhamnosus* GG (2 × 10^10^ CFU/day).	The probiotic intervention did not influence cognitive performance.
Edebol Carlman HMT et al. [28]	RCT	8 weeks	Sweden	22	24.2 ± 3.4	27/73	The subjects received a combination of three probiotic strains—*Lactobacillus helveticus* R0052 (CNCM-I-1722; 2 × 10^9^ CFU), *Lactiplantibacillus plantarum* R1012 (CNCM-I-3736; 8 × 10^8^ CFU), and *Bifidobacterium longum* R0175 (CNCM-I-3470; 7 × 10^7^ CFU)—at a dosage of 3 g per day.	The probiotic intervention did not influence cognitive performance.
Ascone et al. [29]	RCT	4 weeks	Germany	59	27.1 ± 6.7	43/57	The participants received a multi-strain probiotic (Vivomixx^®^) at a daily dosage of 4.4 g.	The administered multi-strain probiotic did not induce any effects on cognition or mental well-being in young, healthy adults.
Cannavale CN et al. [30]	RCT	4 weeks	USA	26	25–45	58/42	Participants underwent testing before and after a 4-week consumption period, which included 8 oz of a dairy-based fermented beverage containing 25–30 billion CFUs of live and active kefir cultures.	The fermented dairy beverage led to enhanced performance in two aspects of relational memory: misplacement (*p* = 0.04) and object-location binding (*p* = 0.03).

BB68S: *Bifidobacterium longum* BB68S; CFU: colony-forming unit; IED: Intra–Extra Dimensional; PDX: polydextrose; RCT: randomized controlled trial; RBANS: Repeatable Battery for the Assessment of Neuropsychological Status; RVP: Rapid Visual Information Processing; WCST: Wisconsin Card Sorting Test.

**Table 3 nutrients-16-00789-t003:** The impacts of probiotic/prebiotic/symbiotic supplementation on cognitive functions in patients with mild cognitive impairment.

Study	Design	Mean Follow-Up	Country	Sample Size	Average Age (Year)	Sex Male/Female (%)	Intervention	Main Results
Aljumaah MR et al. [31]	RCT	3 months	USA	169	64.4 ± 5.5	38/48	The LGG supplementation consisted of two capsules of Culturelle Vegetarian Capsules comprising a blend of 10 billion CFUs of *Lactobacillus rhamnosus* GG and 200 mg of prebiotic inulin derived from chicory root extract.	The reduction in the relative abundance of the Prevotella and Dehalobacterium genera following LGG supplementation in the MCI group showed a correlation with an enhanced cognitive score.
Asaoka D et al. [32]	RCT	24 weeks	Japan	130	77.2 ± 5.8	26/29	The patients received a daily dosage of a probiotic (*B. breve* MCC1274, 2 × 10^10^ CFU/day).	The ADAS-Jcog subscale “orientation” showed significant improvement; MMSE subscales “orientation in time” and “writing” demonstrated significant improvement, specifically in the subgroup with lower baseline MMSE scores (*p* < 0.05).
Xiao J et al. [33]	RCT	16 weeks	Japan	79	61.3 ± 7.7	100% male	The patients received a daily dosage of a probiotic (*B. breve* A1, 2 × 10^10^ CFU/day).	The probiotic group exhibited a significant improvement in RBANS total score (*p* < 0.0001). Notably, there was a substantial enhancement in domain scores, including immediate memory, visuospatial/constructional, and delayed memory (*p* < 0.0001), observed in both intention-to-treat (ITT) analysis and per-protocol (PP) analysis.
Fei Y et al. [34]	RCT	12 weeks	China	42	76.4 ± 9.6	90/10	The group receiving the probiotic received a daily dosage of 2 g of a probiotic blend.	The probiotic group exhibited a notably higher MMSE score (24.75 ± 2.47), and there were significant improvements in attention and calculation (0.90 ± 0.79 vs. 0.65 ± 0.74, *p* < 0.001) and recall scores (1.95 ± 0.76 vs. 0.70 ± 0.47, *p* < 0.001) in comparison to the control group.
Kobayashi Y et al. [35]	RCT	8 weeks	Japan	19	82.5 ± 5.3	2/98	The patients received *B. breve* A1 capsules, each containing more than 1 × 10^10^ CFU (2 × 10^10^ CFU/day).	MMSE scores showed a significant increase during the intervention (+1.7, *p* < 0.01). POMS2 and GSRS scores exhibited significant improvement during the intervention.
Hwang YH et al. [36]	RCT	12 weeks	Korea	100	69.2 ± 7.0	28/72	*Lactobacillus plantarum* C29-fermented soybean (DW2009) 800 mg per day (1 × 10^10^ CFU/day).	The group receiving DW2009 exhibited more significant enhancements in overall cognitive functions (z = 2.36, *p* = 0.02), particularly in the attention domain (z = 2.34, *p* = 0.02).

ADAS-Jcog: Japanese version of Alzheimer’s Disease Assessment Scale; CFU: colony-forming unit; DW2009: Lactobacillus plantarum C29-fermented soybean; GSRS: Gastrointestinal Symptom Rating Scale; ITT: intention-to-treat; LGG: Lactobacillus rhamnosus GG; MCI: mild cognitive impairment; MMSE: Mini-Mental State Examination; POMS2: Profile of Mood States 2nd Edition; RCT: randomized controlled trial; RBANS: Repeatable Battery for the Assessment of Neuropsychological Status.

**Table 4 nutrients-16-00789-t004:** Impacts of probiotic/prebiotic/symbiotic supplementation on cognitive functions in patients with Alzheimer’s disease.

Study	Design	Mean Follow-Up	Country	Sample Size	Average Age (Year)	Sex Male/Female (%)	Intervention	Main Results
Ton AMM et al. [37]	Clinical Trial	90 days	Brazil	13	78.7 ± 3	15/85	The participants received a probiotic fermented milk (4% kefir) supplement at a dosage of 2 mL/kg/day ^1^.	Most patients exhibited a notable improvement in memory, visual–spatial/abstraction abilities, and executive/language functions (*p* < 0.05).
Akhgarjand et al. [38]	RCT	12 weeks	Iran	90	67.9 ± 7.9	33/67	They received probiotic capsules containing *L. rhamnosus* HA (each capsule with 10^15^ CFU probiotics) or probiotic capsules containing *B. longum* R0175 (10^15^ CFU probiotics per capsule) twice daily.	Cognition showed a significant improvement with MMSE (*p* < 0.0001). Post hoc comparisons revealed a notably greater enhancement in the *B. longum* intervention group (4.86, 95% CI: 3.91–5.81; *p* < 0.0001) compared to both the placebo and *L. rhamnosus* intervention groups (4.06, 95% CI: 3.11–5.01; *p* < 0.0001).
Kobayashi Y et al. [39]	RCT	12 weeks	Japan	117	61.5 ± 6.8	49/51	The individuals took two capsules every day, each containing around more than 2.0 × 10^10^ CFU of *B. breve* A1.	In a stratified analysis, a notable distinction emerged between the *B. breve* A1 and placebo groups concerning the ‘immediate memory’ subscale of RBANS and the total MMSE score in participants with a low RBANS total score at the baseline. The scores on the ‘language’ and ‘attention’ subscales showed a significant increase.
Tamtaji et al. [40]	RCT	12 weeks	Iran	79	76.2 ± 8.1	50/50	The patients received selenium (200 μg/day) plus a probiotic containing Lactobacillus acidophilus, Bifidobacterium bifidum, and Bifidobacterium longum (each at 2 × 10^9^ CFU/day).	The combined use of the probiotic and selenium resulted in a significant improvement in the MMSE test (*p* < 0.001). Cognitive functions significantly improved.

RCT: randomized controlled trial; CFU: colony-forming unit; MMSE: Mini-Mental State Examination; RBANS: Repeatable Battery for the Assessment of Neuropsychological Status; ^1^ Probiotic fermented milk containing *Acetobacter aceti*, *Acetobacter* spp., *L. kefiranofaciens*, *L. delbrueckii*, *L. fructivorans*, *L. fermentum*, *Enterococcus faecium*, *Leuconostoc* spp., *Candida famata*, and krusei.

## Data Availability

Data sharing is not applicable to this article as no new data were created or analyzed in this study.

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
