# Peer review of "Exploring the Influence of Gut–Brain Axis Modulation on Cognitive Health: A Comprehensive Review of Prebiotics, Probiotics, and Symbiotics"

_nutrients, 2024, doi:10.3390/nu16060789_

Round 1

Reviewer 1 Report

Comments and Suggestions for Authors

In general terms, the work conducted by Fekete et al. is well-structured and synthesizes in an excellent manner the topic they cover. Limitations of the study and the field are clearly recognized. I only have one minor question/suggestion around the conclusions. The authors defend the need to define a precise dosing regimen for different types and stages of cognitive impairment. Could they be more precise in their recommendations based on the current literature or give a general advice to allow a clinician or a researcher test this type of gut microbiota modulators?

Author Response

Dear Reviewer 1,

Thank you for your valuable feedback. It's worth noting that precise dosing of probiotics for preventing or treating cognitive impairments is still being researched, and definitive guidelines are yet to be established. Recent studies, including our review, suggest that while probiotics may offer potential benefits like reducing inflammation and enhancing cognitive function in neurodegenerative diseases, they are not miracle cures.

In our analysis, a daily intake of 1-2 probiotic blend capsules containing specific strains proved effective „at least 10 billion CFU per day”. However, it's important to emphasize that probiotics should not be seen as standalone treatments for neurodegenerative diseases. They should be part of a comprehensive treatment plan alongside medication, lifestyle changes, and other interventions recommended by healthcare professionals.

While Bifidobacterium and Lactobacillus show promise, further large-scale trials are needed to fully understand their potential benefits and limitations in neurodegenerative diseases. Currently, there's no consensus on the optimal strains and doses for different conditions. Research in this area is ongoing, and definitive outcomes are still pending.

Thank you again for your insightful comments.

Best regards,

Mónika Fekete MD, PhD

Reviewer 2 Report

Comments and Suggestions for Authors

The manuscript proposes a comprehensive systematic review of the literature of the last 5 years on the effect of probiotics and prebiotics on cognitive health.

The authors selected 23 articles covering clinical studies on 837 healthy individuals, 359 patients with MCI and 299 patients with AD.

From the evaluation of the outcomes emerging from the selected articles, probiotics and prebiotics likely have a beneficial effect on cognitive performance thanks to mechanisms of the gut-brain axis.

The manuscript is flowing and well organized. The introduction is satisfying and well-detailed. The tables are detailed and adequately describe the characteristics of the selected studies. The supporting literature is adequate to the topic, particularly as regards the discussion and the possible mechanisms underlying the benefits that have emerged from the use of probiotics and prebiotics.

I have some observations to implement the work which I consider already of good quality.

- The abstract is a little generic and it would be better to give a more detailed explanation of the results that emerged. For example, the are benefits in the case of healthy elderly individuals but not in the case of healthy elderly individuals but with a good level of physical activity. Even among patients with MCI and AD, there seems to be a benefit while in younger and healthier individuals mixed results seem to emerge.

- From line 50 compared to the previous one the discussion changes drastically. I recommend a more homogeneous passage for better reading.

- A more cautious tone could be useful in line 66. The literature supports the theory of counterbalance with pathogens, but the WHO still does not support this hypothesis due to insufficient data. The use of the conditional could tone down.

- Is there a reason of only the last 5 years were selected? Did a previous job cover the previous period? It would be useful to explain this choice

- The methods talk about supplements but some clinical studies used foods containing probiotics. It may be more consistent not to limit the description of the method to dietary supplements only. Furthermore, probiotics and prebiotics are traditionally present in fermented foods and therefore it could be reductive to talk only about food supplements.

10.3389/fnut.2023.1170841

10.1016/j.neubiorev.2020.07.036

 - A hint on the concept of para-probiotic and postbiotic could be useful, which would justify the benefits of fermented foods even after cooking

https://doi.org/10.31083/j.fbe1601003

Author Response

Dear Reviewer 2,

The manuscript proposes a comprehensive systematic review of the literature of the last 5 years on the effect of probiotics and prebiotics on cognitive health. The authors selected 23 articles covering clinical studies on 837 healthy individuals, 359 patients with MCI and 299 patients with AD. From the evaluation of the outcomes emerging from the selected articles, probiotics and prebiotics likely have a beneficial effect on cognitive performance thanks to mechanisms of the gut-brain axis. The manuscript is flowing and well organized. The introduction is satisfying and well-detailed. The tables are detailed and adequately describe the characteristics of the selected studies. The supporting literature is adequate to the topic, particularly as regards the discussion and the possible mechanisms underlying the benefits that have emerged from the use of probiotics and prebiotics. I have some observations to implement the work which I consider already of good quality.

- The abstract is a little generic and it would be better to give a more detailed explanation of the results that emerged. For example, the are benefits in the case of healthy elderly individuals but not in the case of healthy elderly individuals but with a good level of physical activity. Even among patients with MCI and AD, there seems to be a benefit while in younger and healthier individuals mixed results seem to emerge.

Thank you very much for your valuable feedback. We have revised the abstract and marked the additional sentences.

- From line 50 compared to the previous one the discussion changes drastically. I recommend a more homogeneous passage for better reading.

Thank you very much for your valuable comment. We have added an additional sentence to the 50th line, which provides a connection between the two paragraphs.

 - A more cautious tone could be useful in line 66. The literature supports the theory of counterbalance with pathogens, but the WHO still does not support this hypothesis due to insufficient data. The use of the conditional could tone down.

Thank you, We have put this sentence in the conditional mood.

- Is there a reason of only the last 5 years were selected? Did a previous job cover the previous period? It would be useful to explain this choice

This article serves as a complementary piece to our previous Nutrients article, which also spanned 5 years and focused on dietary supplements examining cognitive functions, summarizing the latest research findings. We indicated this in the methodology with the following sentence: "Our current summary article is a continuation of our previous study (17), focusing on specialized dietary supplements in the context of aging." [Reference: Fekete M, Lehoczki A, Tarantini S, Fazekas-Pongor V, Csípő T, Csizmadia Z, Varga JT. Improving Cognitive Function with Nutritional Supplements in Aging: A Comprehensive Narrative Review of Clinical Studies Investigating the Effects of Vitamins, Minerals, Antioxidants, and Other Dietary Supplements. Nutrients. 2023;15(24):5116.]

- The methods talk about supplements but some clinical studies used foods containing probiotics. It may be more consistent not to limit the description of the method to dietary supplements only. Furthermore, probiotics and prebiotics are traditionally present in fermented foods and therefore it could be reductive to talk only about food supplements.

Thank you very much, We have supplemented the methodology with this sentence and marked it in red.

- A hint on the concept of para-probiotic and postbiotic could be useful, which would justify the benefits of fermented foods even after cooking https://doi.org/10.31083/j.fbe1601003

Thank you, We have incorporated this addition into the discussion and highlighted the entire paragraph.

We appreciate your valuable feedback and have made the necessary adjustments accordingly.

Best regards,

Mónika Fekete MD, PhD